# The Effects of Linoleic Acid Consumption on Lipid Risk Markers for Cardiovascular Disease in Healthy Individuals: A Review of Human Intervention Trials

**DOI:** 10.3390/nu12082329

**Published:** 2020-08-04

**Authors:** Erik Froyen, Bonny Burns-Whitmore

**Affiliations:** Department of Nutrition and Food Science, Huntley College of Agriculture, California State Polytechnic University, Pomona, CA 91768, USA; bburnswhitmo@cpp.edu

**Keywords:** cardiovascular disease, linoleic acid, omega-6, polyunsaturated fatty acids, cholesterol, triglycerides, lipoproteins, intervention trial, n-6

## Abstract

Cardiovascular disease (CVD) is the leading cause of death worldwide. Risk factors for developing this disease include high serum concentrations of total cholesterol, triglycerides, low-density lipoproteins, very-low density lipoproteins, and low concentrations of high-density lipoproteins. One proposed dietary strategy for decreasing risk factors involves replacing a portion of dietary saturated fatty acids with mono- and polyunsaturated fatty acids (PUFAs). The essential omega-6 PUFA, linoleic acid (LA), is suggested to decrease the risk for CVD by affecting these lipid risk markers. Reviewing human intervention trials will provide further evidence of the effects of LA consumption on risk factors for CVD. PubMed was used to search for peer-reviewed articles. The purpose of this review was: (1) To summarize human intervention trials that studied the effects of LA consumption on lipid risk markers for CVD in healthy individuals, (2) to provide mechanistic details, and (3) to provide recommendations regarding the consumption of LA to decrease the lipid risk markers for CVD. The results from this review provided evidence that LA consumption decreases CVD lipid risk markers in healthy individuals.

## 1. Introduction

Heart disease is the number one cause of death in the United States, accounting for 647,457 (23%) of the deaths, followed by cancer with 599,108 deaths (21.3%). Stroke is the fifth leading cause of death, contributing to 146,383 deaths (5.2%) in 2017 [1]. Worldwide, heart disease and stroke are the largest contributors to death, accounting for 15.2 million deaths in 2016 [2]. A suggested dietary strategy to decrease the risk factors for cardiovascular disease (CVD) (includes heart disease and stroke) is to decrease the consumption of saturated fatty acids and increase unsaturated fatty acids [3,4,5,6,7]. However, this recommendation has been questioned by various authors/publications [8,9,10,11,12,13]. One such controversy is the recommendation of linoleic acid (LA) intake, an essential omega-6 (or n-6) polyunsaturated fatty acid (PUFA) [14,15,16]. For example, it has been suggested that replacement of saturated fat with LA decreases serum cholesterol, but does not decrease the risk of death from coronary heart disease (CHD) [9]. Additionally, there has been concern that consuming high amounts of LA may increase the risk of inflammation [17].

In a recent analysis of 30 prospective observational studies, higher tissue and circulating concentrations of LA were significantly associated with a decreased risk of cardiovascular events [18]. The Cardiovascular Health Study, a prospective cohort study, found that higher circulating LA (but not other omega-6 fatty acids) decreased total and CHD mortality in older adults [19]. In a prospective cohort study, replacing 5% of energy from saturated fats with PUFAs, was associated with a 25% decreased risk of CHD [20]. In one population-based cohort study, there was an inverse association between total dietary PUFAs, including LA and omega-3 PUFAs, and CVD mortality [21]. Furthermore, according to meta-analyses of prospective cohort studies, a low consumption of n-6 PUFAs and higher intakes of saturated and trans-fatty acids increased CHD mortality [22]. Another meta-analysis of prospective cohort studies indicated that LA consumption decreased the risk of CHD events and deaths, and the authors recommended replacing saturated fat with LA to prevent CHD [23]. In a systematic review of 19 randomized controlled trials in which dietary saturated and monounsaturated fatty acids were replaced with omega-6 fatty acids, Hooper et al. [24] discovered that increasing omega-6 fats decreased the risk for myocardial infarction and that omega-6 fats decreased total serum cholesterol but not “other blood fat fractions”. It was also emphasized that “the benefits of omega-6 fats remain to be proven” [24].

The “diet-heart” hypothesis states that a high consumption of saturated fat and cholesterol and a low consumption of polyunsaturated fat, such as LA, increase the accumulation of cholesterol and plaques in artery walls. These events subsequently lead to the development of atherosclerosis, heart disease, and myocardial infarction [9,10]. However, there are various mechanisms that increase the risk of CHD, such as high blood pressure, thrombosis, arrhythmia, inflammation, endothelial dysfunction, insulin resistance, oxidative stress, cigarette smoking, family history, obesity, and overall dietary patterns, among others [10,25]. Lipid levels are another such mechanism that have been suggested to be “strong” risk factors for CVD and mortality, such as high cholesterol, triglycerides (TGs), low-density lipoprotein cholesterol (LDL-C), lipoprotein(a) (Lp(a)), very-low-density lipoprotein cholesterol (VLDL-C), and low high-density lipoprotein cholesterol (HDL-C) [10,26]. Additionally, apolipoprotein A1 (apoA1; associated with HDL) and apoB (associated with LDL) have also been utilized as risk markers for CVD and mortality [26].

Recent reviews have primarily focused on n-3 PUFAs (such as the essential omega-3 fatty acid alpha-linolenic acid (ALA), eicosapentaenoic acid (EPA), and docosahexaenoic acid (DHA)) and CVD risk, with limited attention to other fatty acids such as LA [27,28,29,30]. Thus, this review focused on the consumption of LA impacting lipid risk factors for CVD in healthy individuals. Specifically, the main objective of this paper was to review human intervention trials that investigated the consumption of LA on lipid levels that have been suggested to affect the risk for CVD, including cholesterol, TGs, LDL-C, VLDL-C, HDL-C, Lp(a), apoA1/2, and apoB. Significant oil and food sources of LA are included in Table 1 and Table 2, respectively. Results from this review will provide human clinical evidence of the effects of LA consumption on CVD risk factors in healthy individuals.

## 2. Materials and Methods

PubMed [32] was utilized to locate peer-reviewed journal articles that investigated the effects of LA consumption on lipid markers of CVD risk in healthy individuals. The search of journal articles occurred during April–June of 2020. The following search terms were used: Linoleic acid, omega-6, n-6, polyunsaturated fatty acids, supplement, lipoprotein, healthy, human clinical trial, randomized controlled trial, VLDL, LDL, HDL, triglycerides, cholesterol, and combinations thereof. Articles were included if they contained information on LA consumption, such as grams or percentage of energy, on lipid biomarkers for CVD risk in healthy individuals. The article was included if the authors stated that the subjects were “healthy”. There were no age restrictions and most of the participants were not obese. Human intervention trials were at least 1 week in length; therefore, postprandial and animal studies were not included. Articles were also excluded if the authors did not state the amount of LA consumed and if it was not indicated whether the subjects were healthy. Using these criteria, 16 journal articles were found.

## 3. Results

The literature search included results from human intervention trials investigating the effects of LA consumption on lipid biomarkers for CVD risk in healthy individuals. The findings are organized by cholesterol, TGs, LDL-C, VLDL-C, HDL-C, Lp(a), and apoA1, apoA2, and apoB. The detailed results are presented in Table 3.

### 3.1. Total Blood Cholesterol

LA intakes decreased total cholesterol concentrations compared with a typical U.S. diet [33], diets high in stearic acid (SA) [34], saturated fatty acids (SFAs) [35], monounsaturated fatty acids (MUFAs) [36], or medium-chain fatty acids (MCFAs) [37]. A higher percentage of energy from LA produced a more significant decrease in cholesterol concentrations [33,36,38], such as compared with palmitic acid (PA) consumption [38]. However, consuming a diet high in dietary cholesterol and LA increased cholesterol concentrations [39]. Goyens et al. [40] demonstrated that a high consumption of alpha-linolenic acid (ALA), along with LA, significantly decreased cholesterol concentrations compared to control with the same percentage of LA. No significant differences in cholesterol were observed after comparing consumption of a diet high in LA with a diet high in EPA and DHA [41], diets rich in oleic acid (OA) [42,43], SA [43], or ALA [44]. Consuming high and low amounts of LA—along with fish oil capsules (rich in EPA and DHA) or olive oil (rich in OA)—were also not different [45]. Furthermore, consumption of high and low amounts of LA with the same amount of ALA (~1% energy (E)) did not impact cholesterol concentrations [46]. In a study by Dias et al. [47], participants consumed a SFA-rich diet or an n-6 PUFA-rich diet (12.7% E LA), both supplemented with EPA and DHA, for 6 weeks, and found no significant differences in cholesterol concentrations.

### 3.2. Triglycerides (TGs)

LA consumption decreased TGs compared with diets rich in SA [34], OA [36], or MCFAs [37]. High and low LA consumption, with fish oil capsules (rich in EPA and DHA), decreased TGs compared with high and low LA consumption with olive oil capsules (rich in OA). Low consumption of LA with fish oil resulted in more of a decrease in TGs compared to high LA consumption and fish oil; however, it was not statistically significant [45]. Interestingly, a SFA-rich diet and an n-6 PUFA-rich diet, both supplemented with EPA and DHA, decreased total TGs; however, there were no significant differences between diets [47]. In contrast, LA increased TGs compared to diets supplemented with fish oil (rich in EPA and DHA) [35,41]. There were no significant differences between lower and higher LA consumption [33,40,46], a diet high in OA versus a diet high in LA [42], an ALA-rich diet versus an LA-rich diet [44], or a diet high in SA versus a diet high in LA [43].

### 3.3. Low-Density Lipoprotein Cholesterol (LDL-C)

Consumption of LA decreased LDL-C, with more significant decreases produced from a higher LA diet [33]. Diets containing LA decreased LDL-C compared with a diet rich in SA [34,48], SFAs [35], trans-fatty acids (TFAs) [48], OA [36], or MCFAs [37]. Gradually increasing dietary LA (up to 10% E) decreased LDL-C, with most significant decreases observed at the highest % E of LA (10%). Palmitic acid (PA) (10% E) remained constant during this study design [38]. In contrast, LA increased LDL-C compared with a diet rich in MUFA (14.5% E OA) [48]. There were no significant differences in LA consumption compared with diets rich in OA [42,43], ALA [44], or SA [43]. Consumption of low and high LA diets, with ALA maintained at about 1% for each diet, did not differ regarding LDL-C concentrations [46]. Furthermore, LDL-C concentrations did not differ following consumption of low and high LA diets—along with fish oil capsules (high in EPA and DHA) or olive oil capsules (rich in OA) [45]. Lastly, consumption of a SFA-rich diet or an n-6 PUFA-rich diet (12.7% E LA)—both supplemented with EPA and DHA—resulted in no differences in LDL-C [47].

### 3.4. Very-Low-Density Lipoprotein Cholesterol (VLDL-C)

VLDL-C concentrations decreased following consumption of a PUFA-rich corn oil (11.3% E LA) compared with a MUFA-rich mixture of olive and sunflower oils (13.6% E OA) [36]. A low consumption of LA (3% E) decreased medium VLDL compared to control diet (7% E LA) [40]. Consumption of a supplement spread containing mostly LA decreased VLDL-C compared with consumption of a supplement spread containing MCFAs [37]. In contrast, there were no significant differences in VLDL-C concentrations after consumption of a SFA-rich diet or an n-6 PUFA-rich diet (12.7% E LA)—both supplemented with EPA and DHA; however, both diets reduced VLDL-C [47].

### 3.5. High-Density Lipoprotein Cholesterol (HDL-C)

LA consumption increased HDL-C compared with a diet high in SA [34]. Interestingly, consumption of LA increased HDL_3_-C and decreased HDL_2_-C compared with an n-3 diet (1.5% E EPA and DHA) [35]. Diets higher in LA decreased HDL-C compared with consumption of a fish oil supplement (rich in EPA and DHA) [41] or a diet rich in PA [38]. There were no significant differences in HDL-C following consumption of higher and lower amounts of LA [33,40,46], a diet high in LA compared with diets rich in OA [36,42,43], ALA [40,44], or SA [43]. Consumption of high and low amounts of LA, with supplements of fish oil capsules (rich in EPA and DHA) or olive oil capsules (rich in OA), did not significantly differ in regards to HDL-C [45]. Furthermore, the consumption of a SFA-rich diet or an n-6 PUFA-rich diet (12.7% E LA)—both supplemented with EPA and DHA—did not significantly differ with respect to HDL-C [47].

### 3.6. Lipoprotein(a) (Lp(a)) Particles

A high consumption of LA decreased Lp(a) compared to a diet high in TFAs [48]. However, consumption of LA (1.5% E) increased Lp(a) compared with a SFA-rich diet [35]. No significant differences were observed when LA consumption (about 12% E) was compared to diets high in SFA, OA, and SA [48].

### 3.7. Apolipoprotein (apo) A1, ApoA2, and ApoB

ApoA1 increased following consumption of both a lower and higher LA diet compared to a typical U.S. diet [33]. Consumption of LA (1.5% E) increased apoA2 compared with an n-3 diet (1.5% E of EPA and DHA) [35]. ApoA1 and apoA2 decreased after an LA-rich diet compared with a diet high in OA [42]. There were no significant differences in apoA1 concentrations following consumption of lower and higher LA diets [40], and consumption of LA compared with diets high in SA or OA [43].

Lower and higher LA consumption decreased apoB compared with a typical U.S. diet, with further decreases generated by the higher LA diet (10.8% E) [33]. An LA-rich diet also decreased apoB compared to diets high in SA, elaidic acid (a TFA) [34], or SFAs [35]. Consumption of a high-ALA diet (7% E LA; 1.1% E ALA) decreased apoB compared to the control diet (7% E LA; 0.4% E ALA) [40]. No significant differences were observed in apoB concentrations after consumption of LA compared to diets rich in OA [42,43] or SA [43].

## 4. Discussion

The major findings from the review will be presented in this section. Relating the results to the proposed mechanisms will also be discussed. Lastly, recommended intakes of LA and future studies will be suggested.

### 4.1. Major Findings of the Reviewed Studies of LA Decreasing the Risk Factors for CVD

This review sought to summarize the results from human intervention trials that investigated the effects of LA consumption on lipid risk markers for CVD. In the review of the literature, it was found that various amounts of LA decreased these risk markers for CVD in healthy individuals. Additionally, consumption of LA decreased total cholesterol concentrations compared with a common U.S. diet [33] and diets rich in SA [34], PA [38], SFAs [35], MUFAs [36], and MCFAs [37], and higher percentages of LA intakes reduced cholesterol concentrations [33,36,38]. LA consumption also reduced TG concentrations compared with diets high in SA [34], OA [36], or MCFAs [37]. Further evidence included intakes of LA decreasing LDL-C compared to diets high in SA [34,48], SFAs [35], TFAs [48], OA [36], or MCFAs [37]. Interestingly, higher amounts of LA decreased LDL-C, while PA consumption remained constant [38]. After consumption of LA, VLDL-C also decreased compared with a diet rich in OA [36] or MCFAs [37]. HDL-C increased following LA consumption compared to a diet high in SA [34]. Furthermore, LA consumption decreased Lp(a) compared to a TFA-rich diet [48] and increased apoA1 compared to a usual U.S. diet [33]. Lastly, apoB decreased after intakes of LA compared to a common U.S. diet [33] and diets rich in SA, elaidic acid (a TFA) [34], or SFAs [35].

### 4.2. Mechanisms by Which PUFAs Decrease Total Cholesterol

According to the aforementioned studies, LA decreased total serum cholesterol compared with other dietary patterns that were not rich in PUFAs [33,34,35,36,37,38]. It was demonstrated that PUFAs increase liver X receptor alpha (LXRα) gene transcription [49,50], perhaps via peroxisome proliferator activated receptors (PPARs) [49]. LXRα increases the expression of cholesterol 7α-hydroxylase (CYP7), which converts cholesterol to bile acids; therefore, by stimulating CYP7 activity, PUFAs help to catabolize cholesterol [51]. The studies that did not observe a significant difference between groups with regards to cholesterol levels primarily involved the additions of n-3 fatty acids (ALA, EPA, and/or DHA)—along with lower and/or higher amounts of LA or n-3 fatty acids as a separate treatment [41,44,45,46,47]. Based on the reviewed studies, LA (or PUFA) consumption decreases total serum cholesterol, which is in agreement with the proposed mechanisms.

### 4.3. Mechanisms by Which PUFAs Decrease Triglycerides

LA consumption decreased TG concentrations compared to a diet rich in MCFAs [37]. On the other hand, there were mixed results with the consumption of LA affecting TG concentrations compared with SA [34,43] or OA [36,42]. The lower % of E (7%) of LA and SA did not differ after consuming each diet for 5 weeks, whereas the higher % of E (~12%) of LA lowered TGs compared to SA after 3 weeks on each diet. Higher consumption of OA (~18% of E) or LA (~18% of E) on TG levels did not differ after 8 weeks on each diet, whereas the intakes of PUFA-rich corn oil (11.3% E of LA and 9.6% E of OA) decreased TG levels compared to consumption of a MUFA-rich mixture (5.7% E of LA and 13.6% E OA) after 2 weeks. The results of these studies (and other reviewed studies) suggest that different treatment durations and percent of E may be factors that lead to different outcomes. 

The consumption of EPA and DHA was more pronounced, which resulted in more significant decreases in TG concentrations [35,41,45]. Furthermore, supplementation with EPA and DHA as part of SFA- and n-6 PUFA-rich diets decreased total TGs (no significant differences between groups) [47]. These results provide further evidence of the TG-lowering properties of EPA and DHA, which coincide with the proposed mechanisms.

Unsaturated fatty acids interact with peroxisome proliferator activated receptor alpha (PPARα), more so than SFAs [52]. PPARα interacts with peroxisome proliferator response elements (PPREs) in the promoter regions of genes, such as apoC-III and lipoprotein lipase (LPL). Certain fatty acids lower TG concentrations by increasing catabolism of lipoproteins and decreasing VLDL secretion from the liver [53]. Fatty acids stimulate LPL, which hydrolyzes TGs from chylomicrons and VLDLs [51,54]. Human studies indicate that dietary omega-3 fatty acids decrease VLDL [55]. It has been suggested that LPL may be “more reactive” to VLDL with PUFA TGs, thereby increasing hydrolysis of TG-rich lipoproteins [51,56].

ApoC-III inhibits LPL activity [57] and, therefore, increases TG concentrations. It is suggested that fatty acids, such as n-3 fatty acids, reduce apoC-III, which increases LPL activity and subsequently VLDL catabolism [51]. It has also been reported that omega-3 PUFAs (including EPA and DHA) decrease TGs [58,59] by decreasing diacylglycerol acyltransferase, fatty acid synthase, and acetyl coenzyme A (CoA) carboxylase [27,60], with DHA possessing more TG-lowering effects [61,62].

### 4.4. Mechanisms by Which PUFAs Decrease LDL-C

The reviewed studies demonstrated that the consumption of LA decreases LDL-C, primarily compared with diets rich in SFAs [34,35,38,48], TFAs [48], or MCFAs [37]. On the other hand, the inclusions of omega-3 fatty acids (ALA, EPA, and/or DHA) with dietary treatments of LA did not differ with respect to LDL-C [44,45,46,47]. These results are in agreement that LA (or PUFA) consumption lowers LDL-C compared to intakes of SFAs or TFAs.

One such mechanism by which SFAs (such as palmitic acid) increase LDL-C is by decreasing LDL receptor protein levels. In contrast, a diet rich in LA has been shown to increase LDL receptor levels [63]. However, it has been suggested that a high consumption of cholesterol decreases LDL receptor messenger ribonucleic acid (mRNA) and, thus, a high consumption will likely not be affected by fatty acid intakes [51]. This is in agreement with one of the reviewed studies whereby a high consumption of cholesterol and LA produced high cholesterol concentrations [39]. SFAs, especially lauric acid (12:0), myristic acid (14:0), and palmitic acid (16:0), decrease LDL receptor activity, protein, and mRNA, whereas unsaturated fatty acids increase these components [51].

PUFAs may exert their LDL-C-lowering properties by increasing membrane fluidity [51,63,64], which increases LDL receptor activity, decreasing LDL apoB and increasing LDL catabolism [65,66,67]. It has been demonstrated that consumption of lauric and myristic acids produce higher LDL-C compared to consumption of long-chain SFAs [65]. Additionally, intakes of lauric, myristic, and palmitic acids increased LDL-C compared to consumption of stearic acid (18:0) by decreasing LDL receptor activity and increasing synthesis of LDL [68]. Interestingly, DHA increases LDL particle size; however, the significance of this outcome needs further clarification [27,30,58,59]. As mentioned previously, PUFAs increase CYP7 activity, thereby converting cholesterol to bile acids in the liver; this mechanism will likely indirectly increase LDL receptor production [51].

It should be noted that the size of LDL particles may differentially affect atherogenesis, as small LDL particles appear to have increased atherogenic effects compared to large LDL particles [13]. It has been suggested that the consumption of SFAs increases large LDL particles. In a recent randomized controlled trial by Bergeron et al. [69], it was demonstrated that higher SFA consumption (~14% E; replaced MUFAs) increased large LDL particles but not medium or small LDLs. These results are in agreement with a previous investigation in which higher SFA intake (18% E) was inversely associated with concentrations of small, dense LDL particles [70].

### 4.5. Mechanisms by Which PUFAs Decrease VLDL-C

PUFAs have been shown to inhibit gene transcription of sterol regulatory element-binding protein-1 (SREBP-1) [71], which is involved in lipogenesis and cholesterol synthesis in the liver [72]. Omega-3 PUFAs, more so than omega-6 PUFAs, seem to possess inhibitory effects on SREBP-1 gene transcription and/or protein [73]. Hence, diets rich in omega-3 PUFAs decrease synthesis of MUFAs, TGs, and cholesterol—in addition to reducing VLDL secretion from the liver [51,73]. It has also been suggested that consumption of n-6 PUFAs increases VLDL catabolism and uptake, which contributes to lower VLDL-C and TG concentrations [27,37]. As previously noted, increased LPL activity, and, therefore, VLDL uptake, are mediated by PPAR via PUFAs [27,52].

These mechanisms add insights as to why intakes of LA decreased VLDL-C compared to diets rich in OA [36] and MCFAs [37], as these are monounsaturated and saturated fatty acids, respectively. Moreover, consumption of SFA- or LA-rich diets (supplemented with EPA and DHA) both lowered VLDL-C [47], as these dietary treatments were rich in n-3 PUFAs (EPA and DHA).

### 4.6. Mechanisms by Which PUFAs Affect HDL-C

HDL particles help to decrease the risk for CVD through a variety of mechanisms, such as removal of cholesterol from macrophages, improvement of endothelial function, and increasing antioxidant and anti-inflammatory properties [74]. Consumption of a Western diet (high fat, high cholesterol) [75] increased oxidized lipids in HDL, such as oxidized arachidonic acid (AA) and LA [76,77,78]. Oxidized fatty acids in lipoproteins have been suggested to increase atherogenesis [79]. As observed in one of the reviewed studies, a high consumption of LA (~18% of E) increased oxidized HDL compared with a diet rich in OA, which is a MUFA [42]. Fish consumption is typically low in a Western diet and, therefore, low in EPA and DHA [30]. Interestingly, it has been reported that EPA lowers HDL_3_-C, whereas DHA increases HDL_2_-C, which is more “cardioprotective” [80,81].

According to a reviewed study, intake of LA increased HDL_3_-C and lowered HDL_2_-C compared to an n-3 diet (1.5% E EPA and DHA) [35]. These results agree with the proposed mechanisms for EPA and DHA. Additionally, diets higher in LA decreased total HDL-C compared with consumption of a fish oil supplement [41]. 

Replacement of SFAs with MUFAs or PUFAs results in lower total cholesterol and LDL-C—along with “slightly” lower HDL-C. However, there is a lower total-C:HDL-C ratio [4,82]. Consumption of LA raised HDL-C compared with a diet high in SA [34]. This suggests that SA is not as potent an inducer of HDL-C as is lauric acid, for example [82]. As expected, higher dietary intakes of LA decreased HDL-C compared with consumption of a diet rich in PA [38]. As previously discussed, there were studies that did not show a significant difference in HDL-C. These investigations used higher and lower amounts of LA [33,40,46], or added n-3 fatty acids to the LA interventions or were compared as separate groups [40,44,45,47]. Furthermore, there were no significant differences comparing LA consumption and diets rich in OA [36,42,43], suggesting that LA and OA have similar effects on HDL-C.

### 4.7. Lipoprotein A as a Risk Factor for CVD

Lp(a), which is made in the liver [83], is a risk factor for CVD (discussed in [84]). Lp(a) contains apolipoprotein (a) (apo(a)), which is bound to apolipoprotein B-100 of “modified” LDL particles [85,86]. Lp(a) differs from LDL, as it also contains apo(a) (discussed in [87]). The biological activity of Lp(a) is not known [87]. Both Lp(a) concentrations and apo(a) size are independent risk factors for CVD [84]. It is unclear if apo(a) size affects atherosclerosis [86]; however, high concentrations of Lp(a) have been suggested to increase the risk for CVD [84,86,88]. Genetics account for more than 90% of Lp(a) concentrations [89]; therefore, concentrations are not typically impacted by diet or exercise [88]. However, it has been reported that elaidic acid (a TFA) increased Lp(a) compared to OA and PA [90]. These results are in agreement with one study whereby consumption of LA reduced Lp(a) compared with a TFA-rich diet [48]. In a previous study, consumption of SA increased Lp(a) compared to PA and lauric + myristic acids [91]. In a reviewed study, consumption of LA increased Lp(a) compared to a SFA diet; however, a low percentage of LA was consumed (1.5% E) [35].

### 4.8. ApoA Is Associated with HDL Particles

ApoA1 and apoA2 are two forms of apoA. HDL particles consist of apoA1, which binds to the Adenosine triphosphate (ATP)-binding cassette transporter (ABCA-1) on cell surfaces. Additionally, apoA1 is a cofactor for lecithin cholesterol acyl transferase, which forms mature HDL (discussed in [26,92]). Plasma apoA1 concentrations are typically related to HDL-C concentrations [26]. The biological activity of apoA2 is less understood [93]. PPARα also interacts with PPREs in the promoter regions of the genes in the liver for apoA1 and apoA2 [53]. Therefore, this provides a mechanism by which PUFAs may influence apoA1 expression [51].

According to the reviewed studies, consumption of LA increased apoA1 compared to a common U.S. diet [33]. However, LA consumption resulted in no significant differences in apoA1 concentrations compared to diets rich in SA or OA [43], which suggests that these fatty acids have a similar impact on apoA1. On the other hand, apoA1 decreased after an LA-rich diet compared with a diet high in OA [42]. As discussed, there are mixed results comparing LA and OA intakes on CVD risk factors.

### 4.9. ApoB and the ApoB/ApoA1 Ratio as Risk Markers for CVD

ApoB also exists in two forms, apoB-48 and apoB-100. The intestine produces apoB-48, which is a component of chylomicrons. The liver generates apoB-100, which is a component of VLDL and LDL particles. Under fasting conditions, >95% of circulating apoB exists as apoB-100. ApoB is needed for the binding of the lipoproteins to the LDL receptor. Plasma concentrations of apoB is “strongly correlated” with LDL-C concentrations (discussed in [26,94]). It has been reported that apoA1, apoB, and the apoB/apoA1 ratio are more associated with reduced risk of CVD and/or mortality than with total cholesterol, HDL-C, and LDL-C as CVD risk markers [26,95,96,97]. An additional study demonstrated that high concentrations of apoB increase the risk for CVD, while apoA1 has a “protective” effect [26,98].

It has been shown that PUFAs increase LDL fractional catabolic rate and thus decrease apoB compared to SFAs [65]. Based on the reviewed studies, LA consumption reduced apoB concentrations compared to a common U.S. diet [33] and diets rich in SA, elaidic acid (a TFA) [34], or SFA [35]. Consumption of higher amounts of ALA, decreased apoB, compared to a control diet with the same percentage of LA [40]. However, there were no significant differences in regards to apoB after comparing LA and OA intakes [42,43]. These results suggest LA and OA similarly impact apoB concentrations.

### 4.10. Linoleic Acid Consumption and Recommended Intakes

In the United States, the usual consumption of LA is about 6% of E [17], with men consuming 16 g/d and women consuming 12.6 g/d, on average [99]. The dietary reference intakes (DRIs), generated by the Institute of Medicine of the National Academies, are provided as adequate intakes (AIs), as not enough scientific information is available to provide recommended dietary allowances (RDAs) for LA. As such, the AIs for men and women (19–50 years) are 17 g/d and 12 g/d of LA, respectively. For men and women ages 51 to 70 years of age, the AIs are 14 g/d and 11 g/d of LA, respectively. The American Heart Association recommends consuming 5 to 10% of E of LA for adults in order to decrease the risk for CHD [15,17]. The World Health Organization recommends 2.5 to 9% of E from LA. It is indicated that the lower percentage prevents deficiency, whereas the higher percentage—along with a “healthy diet”-decreases LDL and total cholesterol levels, and thus decreases the risk for CHD [100]. The amount of LA needed to prevent essential fatty acid deficiency is thought to be 1 to 2% of E [101].

It is suggested that “current data are insufficient” to recommend LA intakes above those that meet the requirements for the essential fatty acid [14]. It is recommended to replace SFAs with MUFAs and/or PUFAs to decrease the risk for CVD. Studies have demonstrated the consumption of both n-6 PUFAs and n-3 PUFAs, such as plant-based ALA and n-3 PUFAs from fish, reduce the risk for CVD. However, the n-6:n-3 ratio is suggested to be a “less relevant indicator” in regards to CVD prevention (discussed in [3,4,10]). Rather, the “Omega-3 Index”, which indicates the EPA + DHA content in red blood cells, is a more useful biomarker, as many individuals consume low amounts of EPA and DHA [102]. The optimum amounts of n-6 and n-3 PUFAs should, therefore, be clarified [10]. Furthermore, it has been stated that there is an “overestimation of benefits, and underestimation of potential risks, of replacing saturated fat with vegetable oils rich in linoleic acid” [9].

There is no upper limit (UL) for LA; however, it is not advised to consume above recommendations due to lack of research studies investigating the consumption of high amounts of LA. It has been suggested that high intakes of LA increase the risk of CVD by LA converting to AA—thereby generating pro-inflammatory eicosanoids (discussed in [17,102]). However, it is thought that consumption of LA “has little effect on tissue” AA and inflammation [3,17,102,103,104]. There is more of a concern of low n-3 PUFAs increasing inflammation [102], as the omega-6 and omega-3 pathways use the same elongation and desaturation enzymes.

### 4.11. Future Research Directions and Insights

It has been stated that the “diet-heart” hypothesis indicates that “intermediate biomarkers” should not be overemphasized, nonrandomized studies should be interpreted with caution, and publications of randomized controlled trials (RCTs) should be timely and complete [9,12]. Concerns for RCTs include monitoring compliance with periodic dietary analyses, measurement errors in dietary assessments (such as food frequency questionnaires, 24-h recalls, and diet records), adding treatments (such as oils) and restricting other dietary components, multiple lifestyle factors influencing risk markers for CVD, smaller number of participants, shorter duration, high-dropout rates, higher cost, and low generalizability, among others [4,8]. Furthermore, it was stated that it will be difficult to determine the effects of LA on the development of atherosclerosis, as it is “confounded” by LA being a replacement for SFAs, TFAs, or n-3 PUFAs [14]. In other words, are there benefits due to increasing LA and/or decreasing another fatty acid? However, it has been proposed that “only RCTs provide sufficient data to evaluate the specific effects of the n-6 PUFA LA” [12]. On the other hand, others feel that “in most situations, large, prospective cohort studies of hard clinical endpoints, when well designed, can provide the best available evidence to inform dietary recommendations” [4]. Nevertheless, both RCTs and prospective observational cohorts are considered the “top two tiers of evidence-based medicine” [12]. However, certain researchers feel that results from RCTs and prospective observational cohorts do not support the recommendation to maintain or increase LA consumption [12].

A recent article by Krauss and Kris-Etherton [105] provided additional insights on future studies that are needed regarding fatty acid intakes on CVD risk including: (1) Interactions of fatty acids with dietary components, (2) racial and ethnic differences, (3) the need for long-term studies, (4) identifying new CVD risk markers, (5) evaluating “dose-response” of fatty acids on CVD risk markers, (6) determining genomic, “epigenomic”, and microbiome influences, and (7) evaluating the effects of individual fatty acids and fatty acid-rich foods.

### 4.12. Translating Fatty Acid Recommendations into Dietary Recommendations

It is generally agreed that the types of fatty acids in the diet, as opposed to total amount, “is the most important feature for reducing cardiovascular risk”, such as the partial replacement of SFAs with unsaturated fatty acids, especially PUFAs, the elimination of TFAs, and consumption of dietary EPA and DHA (rather than fish oil supplements) [4,6,10]. It is also important to emphasize “whole, natural foods and dietary patterns”, as opposed to substitutions of individual nutrients, as there will be interactions of nutrients and bioactive components that will influence CVD risk [27,106]. Translating these fatty acid recommendations into practice includes the consumption of lean meats and poultry, low-fat dairy, vegetable oils, nuts, seeds, and fish. Additional dietary recommendations to decrease the risk for chronic diseases consist of increased intakes of fruits, vegetables, whole grains, and legumes, while lowering the consumption of salt and added sugars [4,6,10].

## 5. Conclusions

The following list describes the main findings from this review:
LA consumption primarily decreased total cholesterol compared with diets that were not rich in PUFAs [33,34,35,36,37,38].LA consumption produced mixed results on TG concentrations [33,34,36,37,40,42,43,44,46], whereas consumption of EPA and DHA more consistently decreased TGs [35,41,45,47].LA (or PUFA) consumption decreased LDL-C compared to diets rich in SFAs [34,35,37,38,48] or TFAs [48]; there were mixed results compared with OA consumption [36,42,43,48].LA consumption decreased VLDL-C compared to diets high in OA [36], or MCFAs [37]. Additionally, intake of a SFA- or LA-rich diet, both supplemented with EPA and DHA, also reduced VLDL-C [47].LA consumption decreased HDL-C compared to PA [38], or EPA and DHA [41]; there were no differences compared to OA intakes [36,42,43].LA consumption decreased Lp (a) compared to a TFA-rich diet [48].LA consumption increased apoA1 compared with a typical U.S. diet [33].LA consumption decreased apoB compared with a typical U.S. diet [33], SFAs [34,35], or TFAs [34]; there were no significant differences between LA and OA intakes [42,43].As mentioned in previous studies, there were no significant differences or mixed results for certain CVD risk markers, notably when comparing LA and OA intakes. These outcomes indicate that additional studies are needed.The recommended intake of LA for optimum health is not known; as such, it is not recommended to consume LA above 10% of E [14,15,17,100].


## Figures and Tables

**Table 1 nutrients-12-02329-t001:** Oil sources of linoleic acid (per 100 g) ^1^.

Oils	Energy (Kcal)	Total Lipid (g)	Linoleic Acid (g)	Alpha-Linolenic Acid (g)	Total Saturated Fat (g)
Canola oil	884	100	18.6	9.14	7.37
Corn oil	900	100	53.5	1.16	13.0
Cottonseed oil	884	100	51.9	0.20	25.9
Grapeseed oil	884	100	69.6	0.10	9.60
Olive oil	884	100	9.76	0.76	13.8
Peanut oil	884	100	32.0	0.00	16.9
Safflower oil	884	100	12.7	0.10	7.54
Sesame oil	884	100	41.3	0.30	14.2
Soybean oil	884	100	51.0	6.79	15.7
Sunflower oil	884	100	65.7	0.00	10.3
Walnut oil	884	100	52.9	10.4	9.10

^1^ Source: U.S. Department of Agriculture, Food Data Central [31].

**Table 2 nutrients-12-02329-t002:** Food sources of linoleic acid (per 1 ounce [28.3495 g]) ^1^.

Food Sources	Energy (Kcal)	Total Lipid (g)	Linoleic Acid (g)	Alpha-Linolenic Acid (g)	Total Saturated Fat (g)
Almonds	164	14.2	3.49	0.001	1.08
Brazil nuts	185	18.8	6.82	0.01	4.52
Cashews	157	12.4	2.21	0.018	2.21
Pecans	196	20.4	5.85	0.28	1.75
Pine nuts	191	19.4	9.4	0.046	1.39
Pistachios	159	12.8	4.0	0.082	1.68
Pumpkin seeds	163	13.9	5.55	0.031	2.42
Sesame seeds	159	13.4	5.78	0.102	1.88
Sunflower seeds	165	14.1	9.29	0.02	1.48
Walnuts	185	18.5	10.8	2.57	1.74

^1^ Source: U.S. Department of Agriculture, Food Data Central [31].

**Table 3 nutrients-12-02329-t003:** A summary of the effects of linoleic acid consumption on lipid risk markers for cardiovascular disease in healthy individuals (ordered by date of publication).

Author (Year)	Study	Subjects	Age	Duration	Treatment	Linoleic Acid Results
Bronsgeest-Schoute et al. (1979) [39]	Randomized crossover	41 total 24 males 17 females	19–35 years mean, 22.1 years	4 weeks	14 to 15% E LA (25 to 50 g/d), with at least 600 mg C for 2 wk, and 2 wk of less than 200 mg C.	↑serum C (high C diet compared with low C diet) ↔TG
Sanders et al. (1983) [41]	Randomized double-blinded crossover	10 total 6 males 4 females	22–35 years	4 weeks	10 g/d fish oil supplement (1.7 g EPA and 1.2 g DHA) or vegetable oil (3.4 g/d LA) for 2 wk.	↔total C, ↑TG, ↓HDL-C (compared with fish oil)
Iacono et al. (1991) [33]	Randomized crossover	11 males	44–62 years mean, 53.6 years	100 days	Baseline period of 20 d (typical US diet, but meeting RDAs), followed by a lower LA diet (3.8% E; 16 g/100 g dietary fat) and a higher LA diet (10.8% E; 37.8 g/100 g dietary fat) for 40 d.	↓total C, ↓LDL-C, ↓apoB (both diets compared to baseline; 10.8% E LA diet had further decreases) ↔HDL-C, ↔TG, ↑apoA1 (both diets compared to baseline)
Zock et al. (1992) [34]	Randomized multiple crossover	56 total 26 males 30 females	Males: 19–48 years mean, 25 years Females: 18–49 years mean, 24 years	9 weeks	Three diets were followed for 3 wk: high in LA (12% E LA), high in SA (11.8% E SA; 3.9% E LA), and high elaidic acid (7.7% E elaidic acid; 3.8% E LA).	↓total C, ↓LDL-C ↑HDL-C, ↑HDL:LDL ratio ↓TG (compared to SA diet) ↓apoB, ↑apoA1:apoB ratio
Mensink et al. (1992) [48]	Intervention trial	58 total 27 males 31 females	Young (specific ages not stated)	53 days	17 d on a diet high in SFA (19.3% E; primarily PA and SA), followed by 36 d on a diet replacing 6.5% total E from SFA with MUFA (15.1% E; 14.5% E OA) or PUFA (12.7% E; 12.6% E LA) diets.	↔Lp(a), ↑LDL-C (compared to MUFA diet)
Mensink et al. (1992) [48]	Intervention trial	56 total 26 males 30 females	Young (specific ages not stated)	3 weeks	SA diet (11.8% E), LA diet (12% E), or TFA diet (7.7% E).	↔Lp(a) (compared to SA diet) ↓Lp(a) (compared to TFA diet) ↓LDL-C
Sola et al. (1997) [42]	Randomized crossover	22 males	Mean, 49.7 (SE, ± 0.6 years)	32 weeks	Stabilization period for 8 wk, followed by two 8 wk dietary treatments separated by an 8 wk washout period. The dietary treatments included a diet rich in OA (18.2% E MUFA) and one rich in LA (18.1% E PUFA).	↑oxidized HDL_3_, ↔total C, ↔LDL-C ↔HDL_2_-C, ↔HDL_3_-C ↔TG, ↔apoB ↓apoA1, ↓apoA2
Sanders et al. (1997) [35]	Randomized crossover	26 males	18–34 years	17 weeks	3 wk of a SFA diet (16% E SFA; mostly PA and SA), followed by 3 wk of an n-3 diet (1.5% E EPA and DHA or 5 g/d) or an n-6 diet (1.5% E LA or 5 g/d), separated by an 8 wk washout period.	↑TG, ↑HDL_3_-C, ↑apoA2 ↓HDL_2_-C (compared with n-3 diet) ↓total C, ↓LDL-C, ↓apoB ↑Lp(a) (compared with SFA diet)
Pang et al. (1998) [44]	Randomized single-blinded	29 males	18–35 years	8 weeks	After a 2 wk stabilization period, the subjects followed either an ALA-rich diet (10.1 g/d, 3.5% E ALA and 12.1 g/d, 3.1% E LA; ALA:LA ratio of 1:0.9) or an LA-rich diet (1 g/d, 0.1% E ALA and 21 g/d, 6.7% E LA; ALA:LA ratio of 1:66) for 6 wk.	↔total C, ↔LDL-C ↔HDL-C, ↔HDL_2_-C ↔HDL_3_-C, ↔TG
Wagner et al. (2001) [36]	Randomized double-blinded crossover	28 males	19–31 years mean, 23.7 years	11 weeks	After 2 wk of adjustment, the subjects consumed 80 g/d PUFA-rich corn oil (11.3% E LA and 9.6% E OA) or 80 g of a MUFA-rich mixture of olive and sunflower oils (5.7% E LA and 13.6% E OA) for 2 wk.	↓total C (after crossover; compared with MUFA-rich mixture) ↓LDL-C, ↓TG, ↓VLDL-TG ↓VLDL-C (after initial 2 wk) ↔HDL-C
French et al. (2002) [38]	Intervention trial	3 males 3 females	Mean, 25 years	8 months	The subjects consumed 8 different diets for 21 d each, with a break of 7 d between diets. The diets provided 10% E PA, with levels of LA starting at 10% E and gradually decreasing to 2.5% E.	↓total C, ↓LDL-C (from 4.5% E to 10% E LA; 10% E LA produced the lowest LDL-C) ↓HDL-C (from 2.5% E to 10% E LA; 2.5% E LA produced the highest HDL-C)
Goyens et al. (2005) [40]	Randomized double-blinded	21 males 33 females	Males: mean, 52.6 years (SD, ± 13.7) Females: mean, 47.7 years (SD, ± 11.1)	10 weeks	Following a 4 wk run-in period, 18 subjects per group consumed a control diet (7% E LA and 0.4% E ALA, ALA:LA ratio of 1:19), low-LA diet (3% E LA, 0.4% ALA) or high-ALA (7% E LA, 1.1% E ALA). Both treatment diets had an ALA:LA ratio of 1:7.	↓total C,↓LDL-C, ↓apoB ↓total:HDL cholesterol ratio (High-ALA group compared with control) ↔HDL-C, ↔apoA1 ↔TG (all groups) ↓medium VLDL (Low-LA group compared with control) ↓small VLDL (High-ALA group compared with control)
Thijssen et al. (2005) [43]	Randomized crossover	18 males 27 females	28-66 years mean, 51 years (SD, ± 10)	17 weeks	Each participant consumed each diet for 5 wk, with a washout period of ≥ 1 wk. The diets did not differ, except for the replacement of 7% E with SA, OA, or LA.	↔total C, ↔LDL-C ↔HDL-C, ↔TG ↔apoA1, ↔apoB ↔total:HDL cholesterol ratio ↔lipoprotein particle sizes (all diets compared)
Liou et al. (2007) [46]	Randomized crossover	22 males	20–45 years mean, 27.9 years (SEM, ± 1.1)	10 weeks	Following a 2 wk phase without consumption of fish and seafood, each subject consumed the high LA diet (10.5% E LA; LA:ALA ratio of ~10:1) and the low LA diet (3.8% E LA; LA:ALA ratio of ~4:1) for 4 wk each. ALA was maintained at ~1% E for each diet.	↔total C, ↔LDL-C ↔HDL-C ↔LDL:HDL cholesterol ratio, ↔TG
Damsgaard et al. (2008) [45]	Randomized double-blinded	64 males	19–40 years	10 weeks	Following a 2 wk run-in period, the participants consumed fish oil capsules (3.1 g/d n-3 LC PUFA; 1.8 g/d EPA, 0.2 g/d DPA, and 1.1 g/d DHA) or olive oil capsules (3.7 g/d OA) for 8 wk. Within each group, the subjects consumed either a low-LA diet (12.7 g LA/100 g fats or 4% E LA) or a high-LA diet (40.3 g LA/100 g fats or 7% E LA; 7.3 g/d higher LA).	↓TG (fish oil groups compared with olive oil groups; TG decreased by 51% in the low-LA group compared to a decrease of 19% in the high-LA group; not significant) ↔total C, ↔LDL-C ↔HDL-C
van Schalkwijk et al. (2014) [37]	Randomized double-blinded crossover	12 males	30–60 years mean, 51 years (SD, ± 7)	12 weeks	A supplement spread of 60 g/d MCFA (65% C8:0 and C10:0) or LCFA (71% LA) for 3 wk, with a washout period of 6 wk between treatments.	↓total C, ↓LDL-C ↓VLDL-C, ↓TG ↓LDL-TG, ↓VLDL-TG
Dias et al. (2017) [47]	Randomized Intervention trial	6 males 20 females	18–65 years	6 weeks	A SFA-rich diet (18.9% E SFA and 2.9% E LA) or an n-6 PUFA-rich diet (12.6% E SFA and 12.7% E LA) for 6 wk. Each diet was supplemented daily with 400 mg EPA + 2000 mg DHA.	↔total C, ↔VLDL-C ↔LDL-C, ↔ HDL-C ↔VLDL-TG, ↔total TG (no difference between diets; however, both diets reduced VLDL-C, VLDL-TG and total TG)

Abbreviations: ALA, alpha-linolenic acid; apo, apolipoprotein; C, cholesterol; d, days; DHA, docosahexaenoic acid; DPA, docosapentaenoic acid; E, energy; EPA, eicosapentaenoic acid; HDL, high-density lipoprotein; LA, linoleic acid; LCFA, long-chain fatty acid; LDL, low-density lipoprotein; Lp, lipoprotein; MCFA, medium-chain fatty acid; MUFA, monounsaturated fatty acid; OA, oleic acid; PA, palmitic acid; PUFA, polyunsaturated fatty acid; SA, stearic acid; SFA, saturated fatty acid; TFA, trans-fatty acid; TG, triglyceride; VLDL, very-low-density lipoprotein; wk, weeks; ↑, increase; ↓, decrease; ↔, no significant difference between groups.

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
