# Peer review of "The Effects of Linoleic Acid Consumption on Lipid Risk Markers for Cardiovascular Disease in Healthy Individuals: A Review of Human Intervention Trials"

_nutrients, 2020, doi:10.3390/nu12082329_

Round 1

Reviewer 1 Report

This is a nice review of human intervention studies examining the effect of linoleic acid (LA) consumption on lipid risk markers for cardiovascular disease in healthy individuals. I have several suggestions for revisions as outlined below.

  1. Abstract: This is a nice explanation of what will be covered in the review, but would it be possible to also provide a brief statement about what the review concluded (i.e., what is stated in section 4.1)?
  2. Lines 17-18: The sentence, “The majority of information from peer reviewed articles was obtained from PubMed” isn’t very specific. Please describe this in more detail, and clarify if this was a systematic review. The description in the Methods suggests that it might have been a systematic review.
  3. Line 37 (and throughout the paper): I would prefer to see the word cholesterol spelled out when it is used alone, and only use the –C abbreviation in places where total, HDL, or LDL cholesterol are being referred to, i.e., total-C, HDL-C, LDL-C. Please check with the journal regarding whether cholesterol should be abbreviated as “C” when used alone.
  4. Lines 73-76: Since you show the oil and food sources for LA in the Tables, I don’t think you need to state the sources with the highest contents within the text.
  5. Tables 1 and 2: Suggest writing the titles as “Oil sources of linoleic acid (per 100 g)” and “Food sources of linoleic acid (per 1 ounce [23.3495 g]).”
  6. Lines 83-84: Was this a systematic review? If yes, please include a description of how many articles were originally located and screened for potential inclusion, whether reference sections of the collected papers were also reviewed for potential additional studies to include. Also, the description of the criteria used to determine whether a study would be included or excluded is rather brief. Were overweight/obese subjects considered healthy, was there an age restriction (no children), etc.? Please expand the inclusion and exclusion criteria, if possible.
  7. Table 3: Were each of the LA results shown as increases or decreases in the table statistically significant? For Goyens et al., it says “both diets had an ALA:LA ratio of 1:7”; there are 3 diets (control, low-LA, and high-ALA), so please clarify “both.” For van Schalkwijk et al., can you provide more details about the amounts indicated as “mainly C8:0 and C10:0” and “mainly LA”. Please put the abbreviations in the footnote into alphabetical order.
  8. Sections 4.2 through 4.6: These sections are appropriately labeled as “Mechanisms by which PUFAS” decrease total cholesterol, triglycerides, LDL-C, and VLDL-C and affect HDL-C. However, since the focus of this paper is intended to be the effects of LA on lipid risk markers, I think in several instances the focus is too heavy on the effects of omega-3, and not as much on omega-6. Please review these sections, and revise as needed to ensure that the potential mechanisms for omega-6 are emphasized (particularly section 4.3).
  9. Lines 267-270: I think more explanation about LDL particle size is needed. You state that the size of LDL particles may differentially affect atherogenesis (small LDL may have increased atherogenicity compared with large LDL particles), followed by a statement that consumption of SFAs typically increases large LDL particles. The reference cited is a “Debates in Nutrition” paper. I think you should cite original studies showing increased atherogenicity of small LDL particles, and provide a more detailed explanation about consumption of SFA (is this as a replacement of CHO, compared to PUFA, etc.).
  10. Lines 273-276: Should this sentence say, “Hence, diets rich in omega-3 PUFAs…”
  11. Lines 342-344: The meaning of the following sentence in not clear: “It has been reported that apoA1, apoB, and the apoB/apoA1 ratio are associated with reduced risk of CVD and/or mortality in comparison with total C, HDL-C, and LDL-C.”
  12. Sections 4:10 through 4:12: These sections contain a lot of quotes. I am not sure not some of the quotations are necessary, e.g., “rational”, “omega”, “better defined”, “healthy eating pattern”, etc. Please review and, if possible, remove some of the quotations, or reword if needed.
  13. Lines 380-383: The potential effect of omega-6 PUFAs on inflammation is a major controversy with their increased consumption. While I understand that the focus of this review is on lipid risk markers, I do think that mention of inflammation is warranted earlier in the paper (i.e., Introduction).
  14. Minor:
    1. Line 59: “endothelial function” should be “endothelial dysfunction”
    2. Line 73: please be consistent with use of upper or lower case “a” in apo
    3. Line 126: “were not significant differences” should be “were no significant differences”
    4. Line 187: “results for human intervention trials” should be “results from human intervention trials”
    5. Lines 217-220: the 1st part of this sentence is a bit awkward in its organization. It seems to suggest that consumption of OA or LA did not differ, when I think the intended meaning is that the effects of OA or LA on TG levels did not differ.
    6. Lines 222: suggest reorganizing the 1st phrase in this sentence to say “The consumption of EPA and DHA was more pronounced…”
    7. Line 265: “significant of” should be “significance of”
    8. Line 300: total C:HDL ratio should be total-C:HDL-C ratio
    9. Line 301: “not as a potent reducer of” should instead be “not as potent a reducer of”

Reviewer 2 Report

Summary

The authors present and interpret a significant amount of human intervention trials on the effects of linoleic acid addition to the diet and its effects on cardiovascular health. The manuscript is well-written, very comprehensive and very polished. It is obvious that a lot of work as been done to review it. I also think it fits well with the topic of the Journal and will be of great interest to its readers.
